# Tensile Strain Capacity Prediction Model of an X80 Pipeline with Improper Transitioning and Undermatched Girth Weld

**DOI:** 10.3390/ma15207134

**Published:** 2022-10-13

**Authors:** Hongyuan Chen, Lianshuang Dai, Heng Xuan, Xiongxiong Gao, Kun Yang, Lei Wang, Qiang Chi, Chunyong Huo

**Affiliations:** 1State Key Laboratory for Performance and Structure Safety of Petroleum Tubular Goods and Equipment Materials, CNPC Tubular Goods Research Institute, Xi’an 710077, China; 2School of Materials Science and Engineering, Xi’an Jiaotong University, Xi’an 710049, China; 3China Oil & Gas Piping Network Corporation, Beijing 100013, China; 4PipeChina Southwest Pipeline Company, Chengdu 610095, China

**Keywords:** strain-based design, weld joint, tensile strain capacity, vintage pipeline, X80

## Abstract

As an important component of strain-based design, the tensile strain capacity (TSC) concept has been extensively used for pipelines that experience expectable plastic strain for both installation and service. However, some stress-based designed pipelines have experienced unforeseen plastic strain in the past decade that resulted in failure. It seems that the tensile strain capacity has gradually become an important requirement for geohazard risk management and pipeline maintenance of stress-based design pipelines. The tensile strain capacity of an X80 pipeline is investigated. The assessment in this work was based on the fracture initiation–control-based limit state. This limit state corresponds to the onset of stable tearing and generally provides a reasonably conservative estimate. Besides that, factors such as wall thickness, material’s strain hardening capacity, toughness, weld strength mismatch, HAZ (heat-affected zone) softening, pipe wall thickness, high–low misalignment, and internal pressure were also investigated to construct a prediction model of the X80 vintage pipeline.

## 1. Introduction

A strain-based approach is the latest progress in finding a solution to the pipeline design in harsh environments. Ductile fracture under axial tensile strain is one of the most severe limit states. For land pipelines, it is always related to the ground movement such as seismic activities, discontinuous frost soil, and mining subsidence. For offshore pipelines, it is always the result of pipeline laying, such as S-lay, J-lay, and reeling lay, which can produce strains of up to 2%, the uneven seabed, etc.

Vintage pipelines account for a large portion of transmission pipelines. In the USA, more than 50% of overall pipeline mileage was built prior to 1950 and more than 75% of overall pipeline mileage was built prior to 1970 [1]. Many vintage pipelines were constructed without the benefits of non-destructive testing (NDT) and the stringent workmanship requirements of today. Prior to 1970, not all girth welds were inspected by NDT methods. Consequently, weld flaws were undetected and left in the completed pipelines. Some of these welds can also have low toughness.

The operational history of the vintage pipelines has demonstrated that the vast majority of the vintage girth welds are safe for their intended service. This safe operation history is, in a major part, attributable to the low stresses experienced by the girth welds of typical buried pipelines. After passing the construction phase, the undetected flaws can remain dormant for decades without any perceived negative consequences. However, in recent years, many vintage pipelines failed under the specified remote strain, which does not exceed the elasticity limit due to their improper transitioning and under-matched girth weld transition [2,3,4]. For these reasons, the Pipeline and Hazardous Materials Safety Administration (PHMSA) of the U.S. Department of Transportation (DOT) advised that operators should give special attention to girth welds with improper transitioning, and variations in wall thickness when located in pipeline segments where significant pipe support and backfill settlement issues after installation may be present [5].

Failures of vintage girth welds are usually associated with additional stresses imposed on the welds beyond the historical norm. For instance, a disturbance to the pipe support conditions subjects the welds to longitudinal loads or bending moments. The disturbance may come from the ground settlement, landslide, soil creep, or nearby construction activities. Another source of additional tensile stress is the thermal stress when certain segments of the pipeline experience colder than usual temperatures due to compressor station bypass, unusually cold temperatures, etc.

In 2020, the Canada Energy Regulator issued a notice about the accident on the high-grade pipeline girth weld [6]. It can be summarized that recent frequent accidents of the girth weld of the high-grade pipeline are often caused by axial strain and the under-match of the girth weld, even if the welds met the requirements of weld procedure specification.

Integrity management of vintage pipelines may involve the identification of girth welds that have an elevated risk of failures due to ground movement events that might occur in the future. In other cases, integrity assessment of girth welds may be necessary after a ground movement event is known to have occurred. Tensile strain capacity assessment is the most appropriate tool for such assessment [7].

In the past few years, various strain-based design models were developed by CRES/PRCI [8,9,10], SINTEF [11,12], ExxonMobil [13,14,15], JFE [16,17], Ghent University [18], and TWI [19]. These models are all specific for the strain-based designed pipeline, so they all assume overmatching of the weld strength and the upper shelf fracture behavior characteristics. Of course, the combination of unequal wall thickness is not considered.

Despite the basic requirements of equal wall thickness, weld strength overmatching, and upper shelf fracture behavior, considerable research beyond these necessary conditions has been done over the past few years. TWI evaluated methods for the treatment of axial misalignment in Failure Assessment Diagram (FAD)-based fracture assessment of pipeline girth welds with regard to whether the local bending stress due to misalignment should be considered primary or secondary [20]. The PRCI study further introduced the effect of misalignment on the tensile strain capacity of the pipeline and developed recommendations on the incorporation of rational limits for misalignment into workmanship criteria and for applications of alternative flaw acceptance criteria [21], such that those in API 1104 Annex A and CSA Z662 Annex K [22,23]. Although the undermatching was not accepted by the strain-based design, the influence on the strain-based design (SBD) was still investigated to some extent [24,25].

In recent years, the most representative research is the test procedure and definition of apparent toughness. Besides the standardization of test methods, the apparent toughness of the CTOD expression has been investigated [26].

A project for tensile strain capacity assessment of in-service vintage pipelines introduced undermatching and low fracture toughness has been launched. However, the project did not induce the improper transition, and the scope of strength and specifications is lower than X65 and smaller than OD1016mm separately [27].

In China, pipelines in harsh environments and geological conditions pose the risk of potential strain-controlled load. Besides that, as many pipelines use a semi-automatic welding process and pipe-bends transition, welds may have low toughness, unequal wall thickness, and strength undermatching. In order to effectively assess the strain capacity of such a pipeline under stratigraphic movement and provide the basis for geohazard management and pipeline quality improvement, it is necessary to assess the pipeline tensile strain capacity under improper transition and undermatching conditions. Based on the former research work in the field, the work introduced the pipeline unequal wall thickness/improper transition, girth weld matching, apparent toughness, and other factors, and developed the tensile strain capacity model for the specific X80 pipeline through the multidimensional grid interpolation method.

## 2. Characteristics and Properties of the Girth Weld

The TSC analysis was performed following the Level 4a procedures of tensile strain models, utilizing advanced finite element analysis (FEA). Values of key input parameters were obtained from the incident information, TGRI lab fabricated welds, or estimated based on the internal database.

### 2.1. Girth Weld Profile and Material Zones

Seven available weld macros were reviewed to identify common characteristics in the weld profile. Figure 1 shows a Flux-Cored Arc Welding Self (FCAW-S) weld macro made on X80 pipes. Meanwhile, Table 1 shows the welding material and the procedures. Based on the weld macro, three material zones were identified: (1) pipe, (2) HAZ, and (3) weld metal. Three zones were used to simplify the modeling of material variation in the weld region while preserving the critical features. Dimensions that characterize the boundaries among the material zones were measured and are listed in Table 2 for each of the seven macros reviewed.

A representative value for each parameter was selected based on the measurements, shown in the last row in Table 2. A weld profile for the FEA was then constructed for each wall thickness combination following the representative values. For transition welds, a taper of 20° was added to the thicker side, as shown in Figure 2. A surface-breaking flaw with 4 mm height and 25 mm length was introduced. The flaw was located at the fusion boundary between HAZ and weld root. When applicable, the flaw was placed at the thinner wall side to obtain a conservative tensile strain capacity (TSC) estimate. An example profile used in the FEA is shown in Figure 2, for the wall thickness combination of 12.8 mm + 15.3 mm.

### 2.2. Material Properties in FEA Models

#### 2.2.1. Linepipe Properties

The pipe nominal size is Φ1016 mm × 12.8 mm, and the specimens are cut from the same pipe. The chemical compositions of the pipe material are listed in Table 3. Three pipe ultimate tensile strength (UTS) levels were selected for FEA, shown in Table 4. The 634 MPa and 772 MPa levels were expected to be reasonably lower and upper bounds for X80 pipes. The 703 MPa level is representative of the pipes involved in the incidents. The yield strength (YS) for each level was then estimated based on an internal database. In general, a higher UTS is correlated with a greater yield-to-tensile (Y/T) ratio.

Pipe stress–strain curves were prepared based on the selected strength combinations following the procedures in CSA Z662 Annex C [20]. Figure 3 shows the constructed pipe stress–strain curves used in the TSC analysis.

#### 2.2.2. Heat-Affected Zone Properties

Two HAZ softening levels, 10% and 20%, were considered in the TSC analysis as typical levels for the weld procedure. Since the material strength in the HAZ is not uniform, the HAZ softening level was defined as the relative difference between the lowest UTS in the HAZ and the pipe UTS outside the HAZ. Figure 4 shows a schematic of the strength distribution modeled in the HAZ.

#### 2.2.3. Deposited Weld Metal Properties

One condition for the deposited weld metal was considered: (1) ER80S root pass with Fabshield^®^ X80 Fill/Cap passes. Representative strengths for ER80S and Fabshield^®^ X80 were estimated based on available data. For both consumables, the YS and UTS were estimated at 552 MPa and 655 MPa, respectively. The same material properties were applied to the entire weld region in the TSC analysis. Figure 5 shows the weld metal stress–strain curve, which is constructed from estimated material YS and UTS following the procedures in CSA Z662 Annex C [20].

The weld CTODA of 0.3 mm, 0.45 mm, 0.6 mm, and 0.75 mm was selected to present the different toughness levels. It covers the main welding processes, including shielded metal arc welding (SMAW), FCAW, gas metal arc welding (GMAW), etc.

## 3. TSC Analysis

### 3.1. Summary of Conditions Analyzed

Table 5 summarizes the conditions analyzed. The considerations to select those conditions have been discussed in previous sections. A total of 48 cases were analyzed.

### 3.2. Finite Element Models

ABAQUS^®^ was used for the FEA. Each finite element model consists of a girth weld with a straight pipe segment on each side. The total model length is six times the pipe OD, with the girth weld at the center. Figure 6 shows an example model used in the FEA.

### 3.3. Analysis Procedure

The FEA consists of two loading steps. In the first step, internal pressure corresponding to a pressure factor of 0.72 was applied. Both ends of the pipe were fixed during this step. In the second step, the internal pressure from the first step was maintained, and a tensile displacement was applied to one end of the pipe while the other end remained fixed. The crack tip opening displacement and the applied strain were recorded to produce a crack-driving force relationship.

The initiation–control-based limit state is achieved when the crack-driving force (CTODF or *δ**F*) reaches the material’s apparent toughness (CTODA or *δ**A*). Both CTODF and CTODA are represented by the crack tip opening displacement (CTOD). The remote strain corresponding to CTODA is the (TSC) in Figure 7. The applied strain was measured using a gauge length of 1.0 pipe OD at a distance of 1.0 OD from the girth weld. This strain measurement is defined as the remote strain, representing the nominal strain in the pipe body under any given loading state.

### 3.4. Summary of TSC under the Assessed Conditions

The TSC for each wall thickness combination is shown in Figure 8, Figure 9, Figure 10 and Figure 11. Within each wall thickness combination, twelve conditions were analyzed. The material parameters for all conditions analyzed are given in the table below the bar charts. For the two transition welds, the remote strain measurements on both sides of the weld are given.

Taking Figure 8 as an example,

A total of twelve conditions were listed, ordered by weld mismatch ratio, HAZ softening, and high–low misalignment.Cases 1–4, 5–8, and 9–12 correspond to the three weld mismatch levels, respectively. Weld mismatch can be seen to have a dominant impact on the TSC of the weld, with low weld strengths significantly reducing the overall TSC.Inside each mismatch group, the first two bars correspond to lower misalignment, and the second two bars correspond to higher misalignment. At higher weld mismatch ratio when the TSC is high, misalignment can be seen to have a significant impact on the TSC. The effect of misalignment is less pronounced when the mismatch ratio is low.At each misalignment level, the two bars represent the 10% and 20% HAZ softening conditions. Similar to the misalignment, the HAZ softening has a greater impact on the TSC when the weld mismatch ratio is relatively high.

The interpretation of the TSC of girth welds joining pipes of different wall thickness, as shown in Figure 8 and Figure 9, is worthy of special attention. When a girth weld with different wall thickness is pulled in the longitudinal direction, more deformation goes to the weaker side than the stronger side. In these cases, more deformation goes to the side with a thinner wall. The thicker side does not go into plastic condition, resulting in strains below 0.25%. The impact on the deformation on the thin side by the various parameters is similar to that of girth welds joining pipes of equal wall and thickness.

When the TSC from girth welds joining pipes of different wall thickness is used in integrity assessment, the strain demand and strain capacity should be compared on the same of girth weld. The side with a thinner wall would have greater strain demand than the thicker side. The high strain demand and TSC on the thin side allow for consistent integrity assessment.

### 3.5. Sensitivity Analysis of Parameters

The sensitivity of parameters like strength mismatch level, internal pressure, apparent toughness, high–low misalignment, unequal thickness transitioning, etc., were analyzed on the basis of the 960 TSC data results.

(1) Strength mismatch level (and high–low misalignment)

As shown in Figure 12, from the perspective of fracture mechanics, both undermatching and high–low alignment may cause a significant increase in the amount of fracture driving force. Combined with the analysis in Figure 13, the overmatching weld joint may have a high TSC. Meanwhile, the effect of the misalignment on the TSC decreased significantly when the weld strength was undermatching. It can be considered that undermatching causes TSC degradation and can cover similar effects of the high–low misalignment. It can be argued that the effect of matching levels on TSC is more advantageous.

(2) Internal pressure

As shown in Figure 14, the internal pressure will affect the fracture driving force of the girth weld flaws. When the internal pressure exceeds 5.6 MPa (design factor 0.5), the fracture driving force increases to about twice that at 0 MPa, but after exceeding 5.6 MPa, the fracture driving force rise slowed significantly. Correspondingly, the tensile strain capacity of the flawed girth weld at 0 MPa internal pressure is about 2 times that of 5.6 MPa internal pressure (design factor 0.5). When the internal pressure exceeds 0.5 times of the specified minimum yield internal pressure, the TSC decline slows down significantly. It can be considered that when the design factor is less than 0.5, the internal pressure affects TSC proportionally. When the design factor is greater than 0.5, the effects of internal pressure on TSC are limited.

(3) HAZ softening

As shown in Figure 15, the tensile strain capacity of the flawed girth weld varies under different heat-affected zone softening (10%, 20%) conditions. However, the HAZ softening will cause the TSC to have certain decline for all apparent toughness and internal pressure levels. Similar to the misalignment, the HAZ softening affects TSC a great deal for overmatching weld, and less for undermatching weld.

(4) Apparent toughness

As shown in Figure 15, the apparent toughness has a stable, near-linear relationship with the TSC under all internal pressure conditions. This near-linear relationship was slightly diminished on the condition of no internal pressure, 10% HAZ softening, and 0.75 mm apparent toughness. However, it may be caused by the grid distortion of the crack tip during the finite element calculation.

(5) Unequal wall thickness weld joint

TSC of unequal thickness girth welds deserves particular attention. When the girth welds with unequal wall thickness are under longitudinal tensile strain, the stress of the thick-wall side will be greater than the other side due to the same axial load with a different cross-sectional area. Therefore, the thin side will be more strained and enters the plastic phase early, resulting in TSC not being higher than 0.4%, even considering the average strain for both sides as a related parameter. The effect of parameters on the strain capacity of a thin side is similar to that of an equal-thickness pipeline.

For TSC assessment of pipeline with unequal wall thickness girth weld, the strain demand and strain capacity should be compared for the same weld. Although the thick side always has a higher strain capacity than the thin side, the thin side always has a higher critical strain value because it adopts the same failure triggering event as the thick side. It generates an illusion that the thin side has a higher strain capacity than the thick side. From the perspective of the overall structure of the pipeline, the average strains for both sides of the girth weld should be considered as the tensile strain capacity of the pipeline with unequal thickness girth welds. Considering the above factors and the TSC results shown in Figure 16, when the strain capacity level is relatively high (such as 0 MPa internal pressure conditions), the unequal wall thickness affects the strain capacity of the girth weld obviously. For example, the strain capacity of 12.8 mm can reach 2%. Meanwhile, for the unequal wall-thickness conditions of 12.8 mm and 15.3 mm, the strain will not exceed 1.14%. Similar to the case of misalignment, when the matching degree is low, the negative effects of unequal wall thickness connections are also relatively weakened due to the low strain capacity level.

### 3.6. TSC Prediction

In this study, combined with the actual working conditions of the pipeline project, the possible nine factors (variables) were set at different levels, and the fracture driving force was calculated. Use the failure criterion to obtain the tensile strain capacity calculation results of the ring welding head. For example, the number of results in Table 5 is 960. Based on the results of the TSC assessment (as shown in Appendix A) within the valid range, here we list the specific method of multivariate grid interpolation:

(1) Some of the parameters were normalized to minimize the effect of unit and value:f_p_ = P_d_D/2t_min_σ_y_   pressure factor(1)
ξ = σ_y_/σ_b_     yield ratio of the pipe body(2)
α = (t_max_ − t_min_)/t_min_   normalized wall thickness difference(3)
η = Δ/t_min_     normalized misalignment(4)
f_p_ is pressure factor, P_d_ is operation pressure, D is diameter, t_min_ is minimum thickness, t_max_ is maximum thickness, ξ is yield ratio of the pipe body, σ_y_ is the yield strength of pipe body, σ_b_ is the tensile strength of pipe body, α is normalized wall thickness difference, η is normalized misalignment, and Δ is misalignment.

(2) For the specific operating condition parameters of the pipeline, it can be interpolated in the linear grid of the results for wall thickness (and their combinations), apparent toughness, wrong margin quantity, HAZ softening degree, welding metal mismatched level, and internal pressure. The multiple linear interpolation function was constructed in the normalized parameter grid space to TSC prediction:TSC = F(f_p_, ξ, α, η, δ_A_, λ)

δ_A_ is the apparent fracture toughness and λ is the HAZ softening level. This function allows the TSC calculation under specific operating parameters. The TSC assessment software (beta version) for the in-service pipeline was also developed with the operating interface shown in Figure 17. Parameters such as unequal wall thickness, misalignment, HAZ softening, weld metal strength mismatch, operating pressure, and apparent toughness are introduced, and each parameter can be input within its specific effective range. The apparent toughness level can be obtained through the conversion relationship of impact toughness or fracture toughness. Through the input of the above parameters, the TSC of the in-service pipelines can be effectively evaluated from an engineering perspective. It provides a convenient and practical assessment tool for in-service pipelines to withstand the strain caused by geological disasters.

## 4. Conclusions

The TSC results highlight desirable characteristics in girth welds that can lead to good TSC performance, namely (1) high weld mismatch ratio, (2) low high–low misalignment, and (3) low HAZ softening. Industry experience has shown that a sensible target TSC, including appropriate safety factors, is approximately 0.75%. For a girth weld with wall thickness transition, strain demand is not uniform on both sides of the weld. The target TSC of 0.75% is applicable to the thin wall side. The following recommendations can be drawn based on such a target:

(1) The weld mismatch ratio should be kept above 1.0. In other words, the weld strength should be not lower than the pipe strength. A high mismatch ratio can be achieved through either higher-grade welding consumables or lower-strength pipes. Either option may also come with practical limits and implications on cost and welding productivity.

(2) High–low misalignment should be kept to a minimum. At the maximum permissible misalignment level (2.2 mm), the TSC could fall under the 0.75% target. At the same time, it is recognized that controlling misalignment in large-diameter pipes at tie-in locations can be very difficult. If a tight limit on high–low misalignment proves impractical, achieving the TSC target would likely hinge on having weld strength overmatching and controlling the HAZ softening (as below).

(3) An ideal target for the maximum HAZ softening level is 10%. At this level, limitations on the high–low misalignment can be somewhat relaxed. Greater levels of HAZ softening may still result in sufficient TSC, provided that high–low misalignment can be managed.

Controlling the HAZ softening level largely comes down to managing the chemical composition (carbon equivalent) of the pipe and the heat input during welding. Recommendations are developed on pipe chemical composition limits and welding procedure selection and qualification requirements targeting primarily X70 pipes. These recommendations are broadly applicable to X80 pipes as well.

A similar TSC target can be used for ranking known geohazard locations for assessments and remedial actions. A key challenge in such a process is obtaining reliable material data on existing assets. The TSC results from this work are expected to provide a reasonably wide range of material characteristics, and, if needed, can be used to establish a lower-bound TSC estimate for locations where material data were not available. However, using a lower-bound TSC could lead to a large number of sites being identified for remedial actions. As a long-term solution, it is recommended that a continuous testing program be established to accumulate material data on existing assets.

## Figures and Tables

**Figure 1 materials-15-07134-f001:**
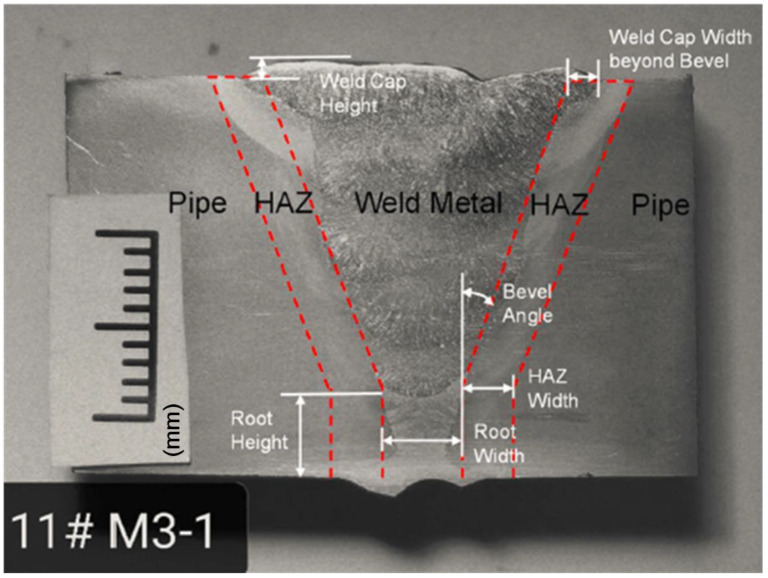
Weld profile measured from the macro of GW 1.

**Figure 2 materials-15-07134-f002:**
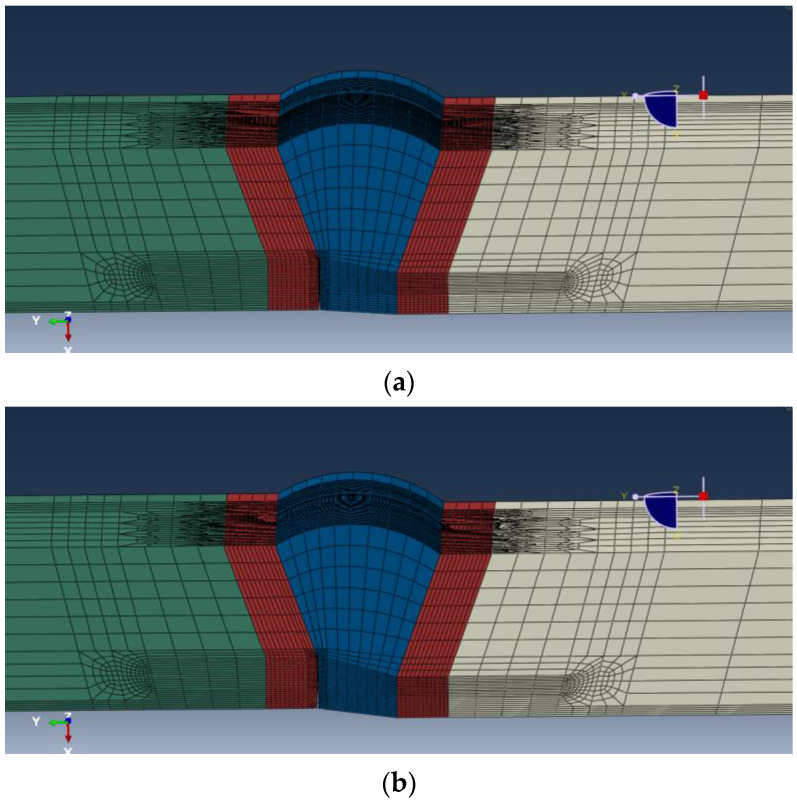
Weld profile modeled in FEA for 12.8 mm + 15.3 mm wall thickness (WT). (**a**) 0.8 mm high–low misalignment; (**b**) 1.6 mm high–low misalignment.

**Figure 3 materials-15-07134-f003:**
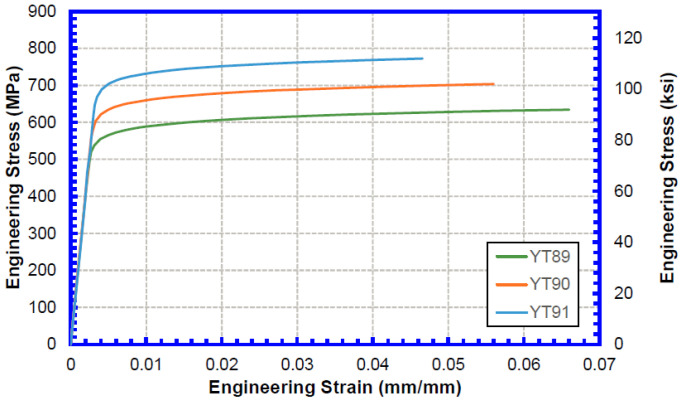
Engineering stress–strain curves for pipe material considered in the TSC analysis.

**Figure 4 materials-15-07134-f004:**
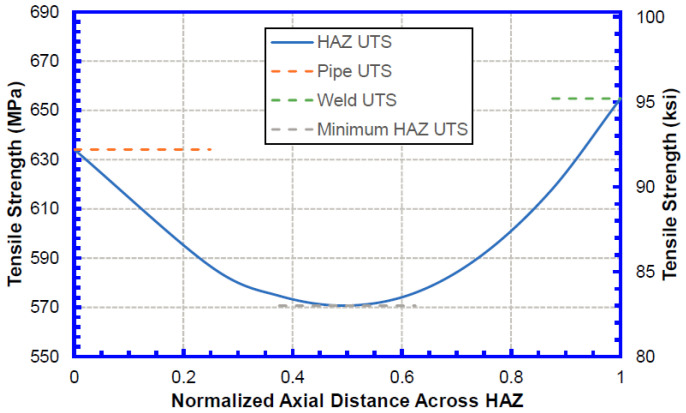
Profile of strength distribution in the HAZ.

**Figure 5 materials-15-07134-f005:**
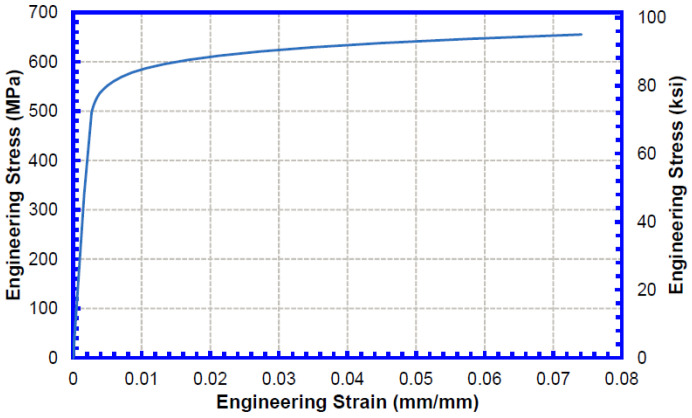
Stress–strain curves of weld metal.

**Figure 6 materials-15-07134-f006:**
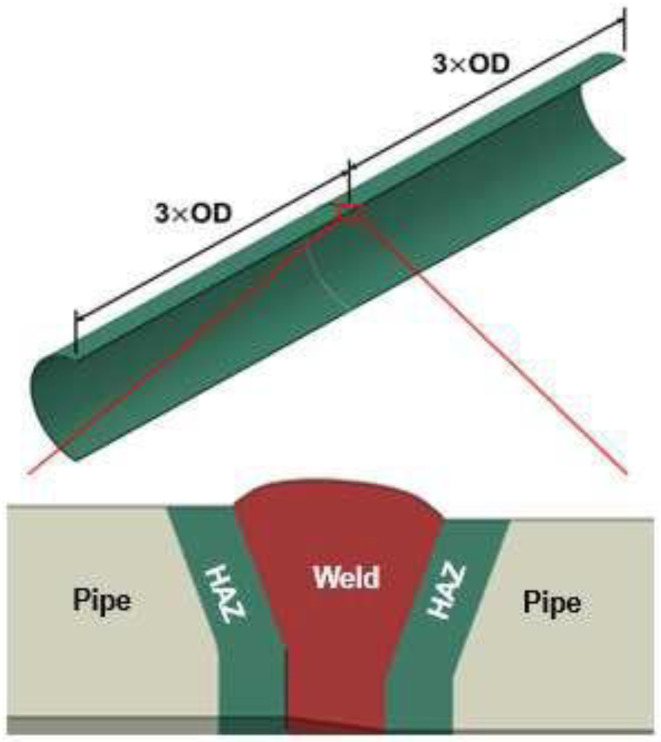
A typical FE model used in the analysis.

**Figure 7 materials-15-07134-f007:**
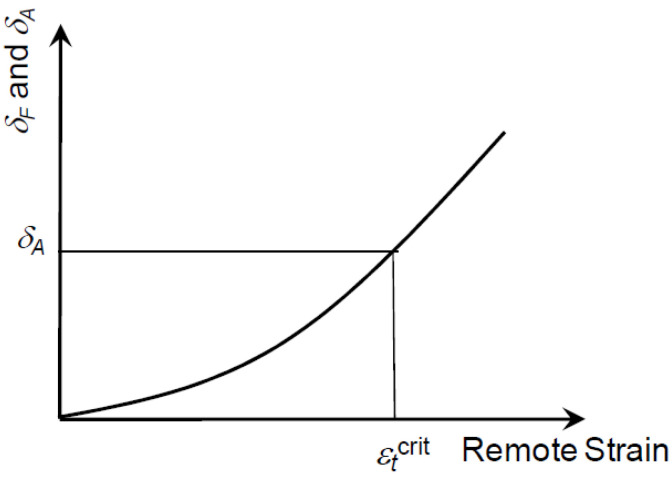
Schematic illustration of a crack-driving force relation.

**Figure 8 materials-15-07134-f008:**
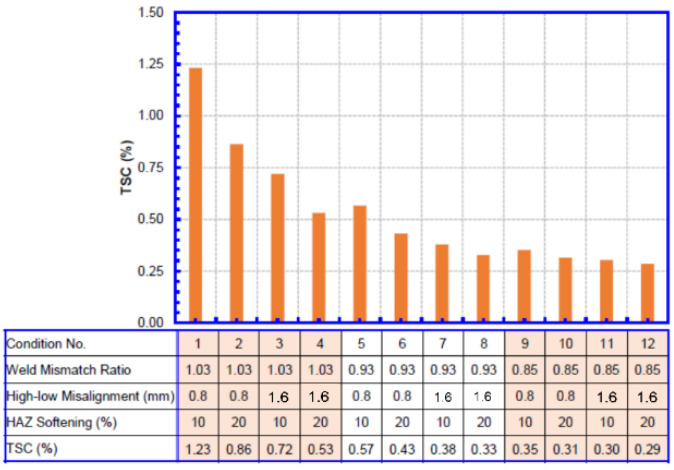
TSC of girth welds with 18.4 mm + 18.4 mm WT.

**Figure 9 materials-15-07134-f009:**
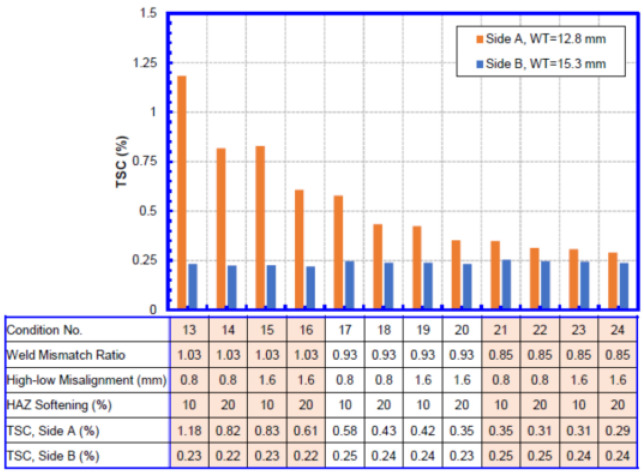
Average TSC of girth welds with 12.8 mm + 15.3 mm WT.

**Figure 10 materials-15-07134-f010:**
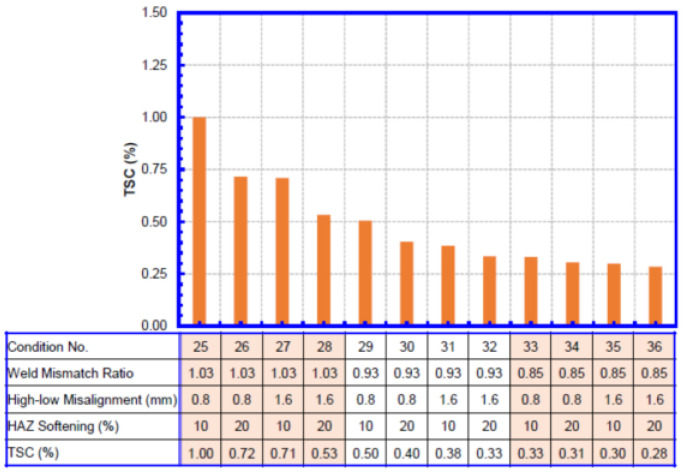
TSC of girth welds with 12.8 mm + 12.8 mm WT.

**Figure 11 materials-15-07134-f011:**
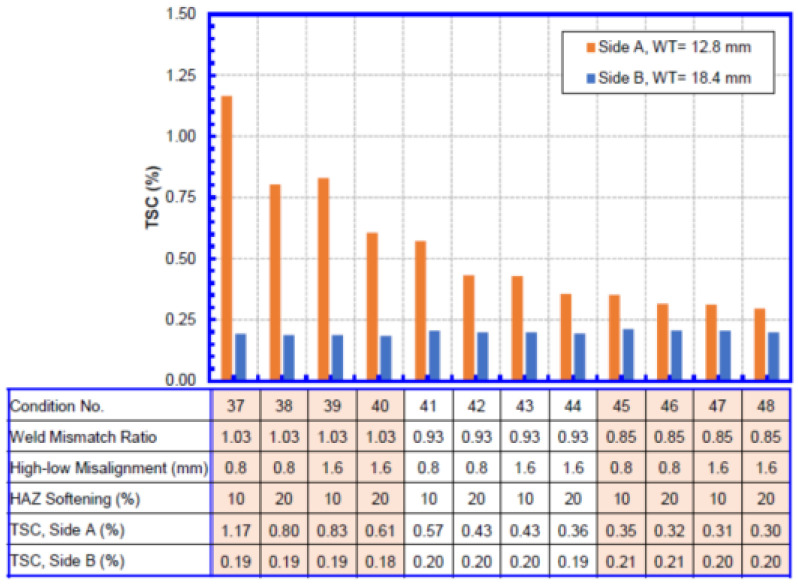
Average TSC of girth welds with 12.8 mm + 18.4 mm WT.

**Figure 12 materials-15-07134-f012:**
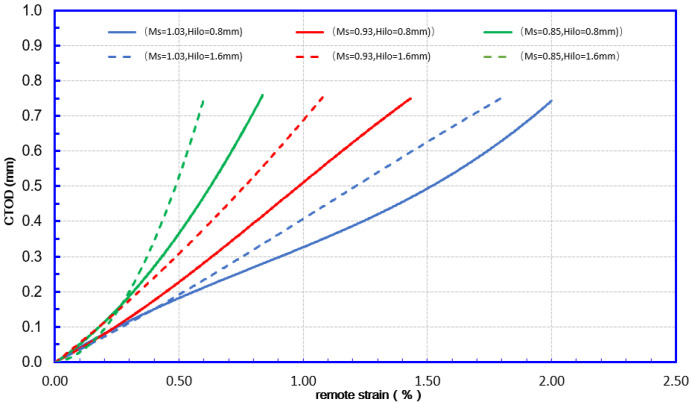
Mismatch effects on CDF 12.8 mm—12.8 mm WT.

**Figure 13 materials-15-07134-f013:**
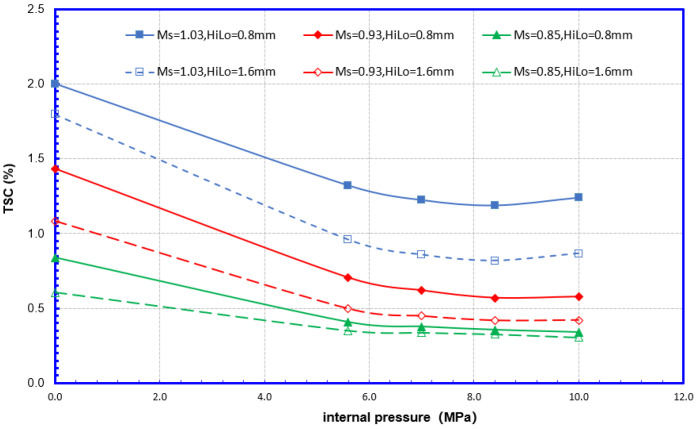
Misalignment and mismatch effects on TSC 12.8 mm—12.8 mm WT.

**Figure 14 materials-15-07134-f014:**
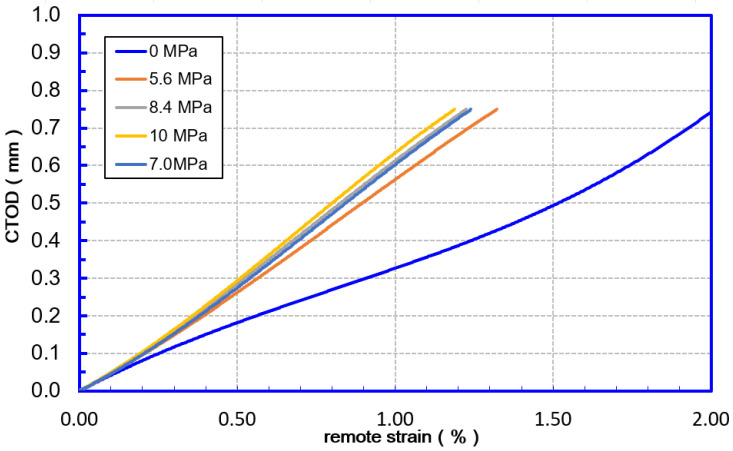
Internal pressure effects on CDF 12.8 mm—12.8 mm WT.

**Figure 15 materials-15-07134-f015:**
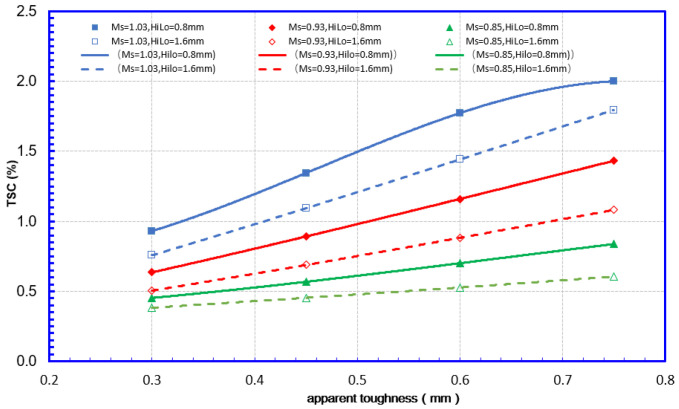
Mismatch and toughness effects on TSC 12.8 mm—12.8 mm WT.

**Figure 16 materials-15-07134-f016:**
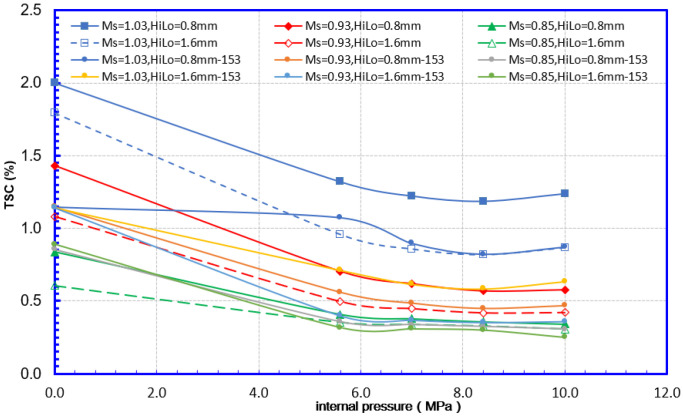
Mismatch and internal pressure effects on TSC 12.8 mm—12.8 mm WT and 12.8 mm—15.3 mm WT.

**Figure 17 materials-15-07134-f017:**
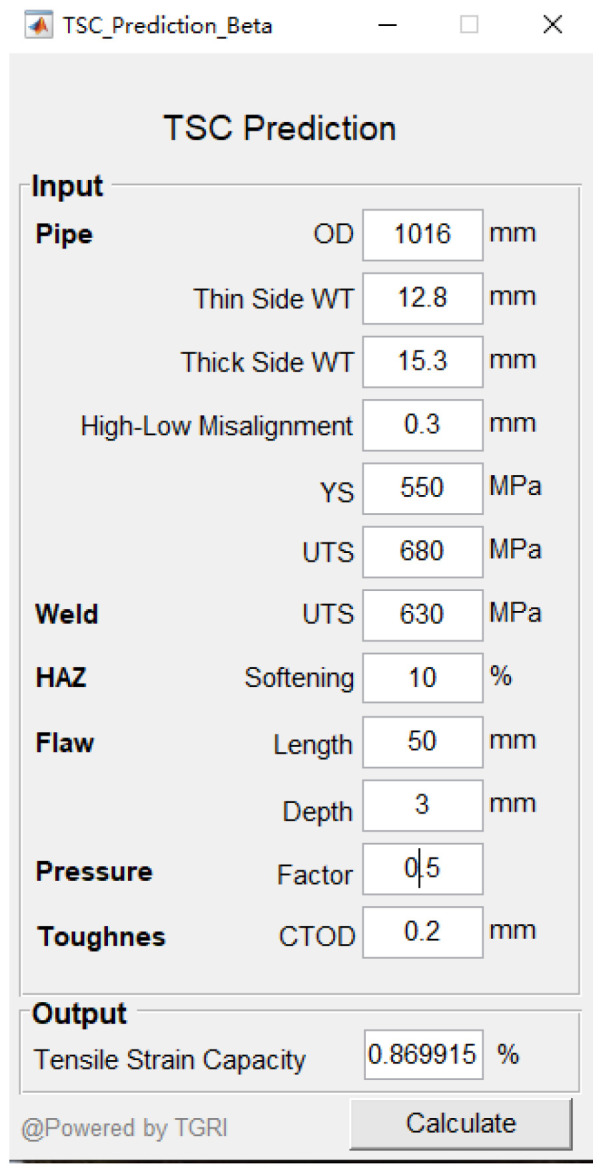
TSC assessment software.

**Table 1 materials-15-07134-t001:** Welding material and the procedures.

Welding Pass	Root Pass	Hot Pass	Fill Pass	Cap Pass
Type	AWS A5.1 E6010	AWS A5.1 E6010	AWS A5.29 E81T8-Ni2	AWS A5.29 E81T8-Ni2
Brand	BOHLER FOX CEL	BOHLER FOX CEL	Golden bridge JC30	Golden bridge JC30
Size	Φ4.0 mm	Φ4.0 mm	Φ2.0 mm	Φ2.0 mm
Welding position	5G	5G	5G	5G
Welding procedure	SMAW	SMAW	FCAW-S	FCAW-S

**Table 2 materials-15-07134-t002:** Longitudinal tensile properties of the pipe.

Position	Bevel Angle (Degree)	Weld Root Width (mm)	Weld Root Height (mm)	Weld Cap Width Beyond Bevel (mm)	Weld Cap Height (mm)	HAZ Width (mm)
GW1	20	4.2	4.6	1.4	1.0	2.7
GW2	23	7.3	3.6	0.0	1.1	2.0
GW3	16	5.6	1.9	2.8	3.0	2.1
GW4	18	6.3	3.5	0.7	2.9	2.4
GW5	20	3.2	3.4	0.0	1.6	2.2
GW6	18	7.3	1.4	1.0	3.3	1.9
GW7	26	5.6	2.8	0.0	1.8	4.0
FEA Model	20	5.6	3.2	0.0	2.0	4.0

**Table 3 materials-15-07134-t003:** Chemical compositions of the pipe material.

C	Si	Mn	P	S	Mo
0.052	0.13	1.49	0.0079	0.0021	0.17
Cr	Nb	V	Ni	Cu	Pcm
0.036	0.051	0.0041	0.17	0.031	0.15

**Table 4 materials-15-07134-t004:** Pipe strengths considered in the TSC analysis.

Pipe YS (MPa)	Pipe UTS (MPa)	Pipe Y/T Ratio
565	634	0.89
634	703	0.90
703	772	0.91

**Table 5 materials-15-07134-t005:** Summary of conditions analyzed.

Parameter	Conditions	Number of Conditions
Pipe	OD 1016 mm	1
Pipe WT	18.4 mm + 18.4 mm 12.8 mm + 12.8 mm 12.8 mm + 15.3 mm 12.8 mm + 18.4 mm	4
Weld profiles	See Table 2	1
Weld CTOD_A_	0.3, 0.45, 0.6, 0.75	4
High–low misalignment	1.6 mm or 0.8 mm	2
Flaw size	25 mm × 4 mm	1
HAZ softening	10%, 20%	2
Weld metal mismatchratio	0.85, 0.93, 1.03	3
Pressure	0 MPa, 5.6 MPa, 7.0 MPa, 8.4 MPa, 10.0 MPa	5
Number of results		960

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
