# Peer review of "Tensile Strain Capacity Prediction Model of an X80 Pipeline with Improper Transitioning and Undermatched Girth Weld"

_materials, 2022, doi:10.3390/ma15207134_

Round 1

Reviewer 1 Report

Title: Tensile strain capacity prediction model of an X80 pipeline with improper transitioning and undermatched girth weld.

Authors: Hongyuan Chen et al…..

Journal: materials, 10.3390

Major corrections:

Line 18: …. Based on the initiation- control based…; specify what is initiated?

Line 72 and 73…. Fracture behaviour. Sentence is incomplete

Line 99 Insert reference for the last line of Page 2 continuing in page 3

As many readers use MPa, include MPa unit along with the ksi

Section 2.2.2 is repeated twice. Correction required

Page 193: … to CTODA is the …. What do you mean by this?

Line 309 – 310: Change the voice in this line.

Line 310: Table 6.3. Where is the table?

Line 310-311: are they single lines. At present separate lines?

Bring some references and discuss salient features by bringing into this paper.

Grammar corrections:

Line 17 ….X80 pipeline is researched (may be corrected as investigated)

Line 54…. Tensile stress not stresses

Line 91 – line starting from And. -such start is not recommended

Page 110, 137, 138 and other lines- Table not table

Page 111, Three zones not the three zones

Page 129, 123, 242: Figure not figure

Page 132 remove bracket close

There are many grammar related errors throughout.

Reviewer 3 Report

The paper is written at a high level.

For a better understanding of the material, I ask you to make several corrections.

Figure 1 shows a scale bar, but does not indicate its size on it. You also made samples, one of which is shown in the indicated figure. However, the procedure does not describe what welding modes this sample was obtained in, what electrodes were used, nor does it describe how the sample was etched and what microscope was used to study the microstructure.

In Figures 12–16, axis labels and text designations are small and hard to see.

In the second output, you write about high and low offset. It would be nice to indicate the percentage or numerical values of the concepts of high and low bias.

Round 2

Reviewer 2 Report

The authors have attended to the recommendations, and, in my opinion, the paper is suitable for publication after minor changes. Please carefully revise the size and format of figures, text font and size in figures, citing, and referencing style according to the journal's specifications.

Author Response

The size and format of figures, text font and size in figures, citing, and referencing style have been revised.

Reviewer 3 Report

In general, my comments were taken into account by the authors and corrected. In my opinion the article can be published.

Author Response

(The authors gave the same response as above.)
